# Secure base effect in former shelter dogs and other family dogs: Strangers do not provide security in a problem-solving task

Giulia Cimarelli[1,2]*, Julia Schindlbauer[1], Teresa Pegger[1], Verena Wesian[1], Zsófia Virányi[1]

**1** Clever Dog Lab, Comparative Cognition, Messerli Research Institute, University of Veterinary Medicine Vienna, Medical University of Vienna, University of Vienna, Vienna, Austria, **2** Domestication Lab, Konrad Lorenz Institute of Ethology, University of Veterinary Medicine, Vienna, Vienna, Austria

* giulia.cimarelli@vetmeduni.ac.at

**Data Availability Statement:** All relevant data are within the paper and its Supporting information files.

## Abstract

Domestic dogs display behavioural patterns towards their owners that fulfil the four criteria of attachment. As such, they use their owners as a secure base, exploring the environment and manipulating objects more when accompanied by their owners than when alone. Although there are some indications that owners serve as a better secure base than other human beings, the evidence regarding a strong owner-stranger differentiation in a manipulative context is not straightforward. In the present study, we conducted two experiments in which pet dogs were tested in an object-manipulation task in the presence of the owner and of a stranger, varying how the human partner would behave (i.e. remaining silent or encouraging the dog, Experiment 1), and when alone (Experiment 2). Further, to gain a better insight into the mechanisms behind a potential owner-stranger differentiation, we investigated the effect of dogs' previous life history (i.e. having lived in a shelter or having lived in the same household since puppyhood). Overall, we found that strangers do not provide a secure base effect and that former shelter dogs show a stronger owner-stranger differentiation than other family dogs. As former shelter dogs show more behavioural signs correlated with anxiety towards the novel environment and the stranger, we concluded that having been re-homed does not necessarily affect the likelihood of forming a secure bond with the new owner but might have an impact on how dogs interact with novel stimuli, including unfamiliar humans. These results confirm the owner's unique role in providing security to their dogs and have practical implications for the bond formation in pet dogs with a past in a shelter.

## Introduction

Caregivers do not only provide infants with protection and support but also with security to explore the environment and interact with different stimuli [1]. This so-called "secure base" effect allows infants to learn about the environment and, ultimately, to become self-reliant,

**Funding:** Writing was supported by the Austrian Science Fund (FWF project P-34749 to GC).

**Competing interests:** The authors have declared that no competing interests exist.

independent adults [1, 2]. Most research aiming at understanding the development and consequences of attachment bonds has been conducted on humans, even though the general theoretical framework was originally developed to explain infants-mother relationships in a variety of species [2]. Over the past 20 years, however, another species has become central in the study of attachment bonds: the domestic dog [3]. Due to its long domestication history and co-existence with humans [4], the dog-human relationship has been suggested to parallel the infant-mother bond and to inform us about its evolution and development [5]. Dogs seem to fulfil all four criteria distinguishing attachment bonds from other affectional bonds (see [6] for a recent review): (a) dogs seek for proximity to their caregiver in novel situations, (b) show distress when separated from their owner, (c) approach the owner in threatening or stressful situations (the so-called "safe haven effect"), and (d) explore the environment more in presence of their owner than when alone (the so-called "secure base effect", [1]).

Owners' secure base effect on their dogs has been shown mainly using a modified version of an experimental task originally designed to measure toddlers' attachment towards their caregivers: the Ainsworth's Strange Situation Test (SST, [1]). In this task, consisting of a series of separations and reunions from/with the caregiver and an unfamiliar person, dogs play and explore more in presence of the owner than when alone or in presence of an unfamiliar experimenter [3, 7–9]. These differing behaviours in presence of the owner vs. a stranger have been interpreted as a sign that owners can provide their dogs with a secure base and that the dog-owner bond is an individualized relationship that makes the caregiver uninterchangeable with others [1].

This means that a dog-owner relationship can be identified as attachment only if the dog can benefit from the presence of its owner more than that of a stranger or even of another familiar being. It is important to realise, however, that such an owner-stranger differentiation depends not only on the strength of the dog's bond to its owners but also on the ease with which a dog accepts strangers as social partners. Stranger acceptance varies greatly across individual dogs and does so independent of the dogs' bonds to their owners [3, 10–12]. Many studies, however, do not make an effort to disentangle these two factors, and attempt to characterize the dog-owner relationship based on the dogs' differing behaviours towards the owner and a stranger. For instance, in a study investigating the development of attachment of guide dogs to their three subsequent caregivers (i.e. the puppy walker, the trainer and, finally, the blind owner), the authors found that young guide dogs discriminate very little between a stranger and their first two attachment figures in the SST [13]. Interpreting these findings is difficult, however, because this lack of differentiation may be due either to the intensive socialization of these dogs and their consequent ability to use strangers as a source of comfort when they are separated from their owner [13] or to their weaker bonds to their initial two, inevitably temporary, caregivers. The attachment of these dogs to their final owner becomes apparent, however, as at a later age they show clearly differing behaviours to their blind owner and strangers [13]. This later finding in guide dogs has been confirmed by other studies that have shown that also pet dogs, after having moved from one household to another, can form strong attachment bonds with their new caregivers [14]. In fact, re-homed adult dogs do not differ in their attachment behaviours when tested with their old or new caregivers [14], suggesting that breaking bonds even repeatedly is not detrimental for dogs to form new attachment relationships even when adults.

Being relinquished in a shelter and adopted later on by a new owner may have different consequences, however [6, 15], if being abandoned by the primary caregiver affects later relationships of dogs, as it does in human children [16]. In dogs, only one non peer-reviewed study has compared the animals' attachment to their caregivers between dogs adopted from a shelter or having lived in the same household since puppyhood [6, 17]. The authors found that

former shelter dogs tested in the SST generally play less, look more towards the partner, and engage more in locomotion (a potential sign of discomfort) than other family dogs [6, 17]. Dogs showed these behavioural patterns both with the owner and with the stranger, suggesting a weak owner-stranger differentiation. Nevertheless, former shelter dogs did not differ from other family dogs in terms of exploration, proximity seeking, physical contact, vocalizations during separation, and greeting. Based on these two sets of results, the authors suggested that former shelter dogs might be more anxious in response to a novel environment and/or to a stranger (as children, [18]), but also that they can indeed form a strong bond with their new owners, even if probably in a less secure way than other family dogs [6, 17]. Based on the fact that former shelter dogs seemed to show a less pronounced owner-stranger differentiation than other family dogs, the authors raised the possibility that former shelter dogs might show a more sociable behaviour towards strangers, as this might have improved their condition in the shelter (i.e. through a favoured treatment by shelter staff) or have favoured their adoption [20]. Having lived in a shelter might affect the dogs' stranger acceptance as well as their bond towards their new owners, in comparison to dogs who have lived in the same family since puppyhood, for a variety of reasons. For instance, in addition to having been abandoned, living in a stressful environment may also represent a traumatic experience [19–21]. Furthermore, owners who end up to relinquish their dog in a shelter may be more likely to have a poor relationship to their dogs [22] or to even mistreat them. To the best of our knowledge, no study has specifically investigated whether dogs with and without a past in a shelter differ in terms of owner-stranger differentiation, specifically regarding the secure base effect. Understanding whether former shelter dogs develop a secure attachment bond with their new caregivers and how the presence of a stranger affects them in potentially challenging situations has both theoretical and practical relevance. In particular, such an investigation could shed light on how flexible dogs are in forming new bonds [15, 23], how novel individualized relationships develop, and what impact the potential disruption of an attachment bond have on cognitive performance [24]. Therefore, in the current study we aimed to directly compare whether and to what extent former shelter dogs and dogs that always lived in the same family (that we refer to as constant family dogs from now on) can benefit from the presence of their owner or of a stranger, as compared to being alone, in a problem solving situation, where one would expect more object manipulation if the owners have a secure base effect on their dogs.

To compare dogs' behaviours in presence of the owner to when in presence of a stranger or when alone, and investigate potential differences between former shelter dogs and family dogs acquired since puppyhood in this regard, we adopted a recently developed experimental paradigm specifically designed to investigate whether owners and only owners have a secure base effect on dogs [25]. The test consists of an object-manipulation task where dogs are free to interact with an object either alone, in presence of their owner, or in presence of a stranger (in a counterbalanced order), with the aim of investigating whether also strangers could provide security to dogs in a problem-solving task [25]. However, in the first study using this paradigm [25], some methodological limitations did not allow to fully compare to what extent owners and strangers could serve as a secure base for dogs. Specifically, while there was a condition in which the owner would be encouraging the dog, no corresponding condition existed for the stranger. Although results showed that dogs spent more time manipulating the object when the owner was encouraging them as compared to when the stranger was silent, and no differences were found between the silent owner and the silent stranger conditions, the possibility is still open that also an encouraging unfamiliar human could facilitate dogs' manipulation, when compared to the owner. Moreover, the presence of a coordinating experimenter (in addition to the owner or the stranger) seemed to have provided some support to the dog even if this was not strictly considered as a partner in the experimental design [25].

To overcome these limitations, in the present study, we adopted Horn et al [25] methodology by using a complete 2x2 design in which both the owner and the stranger remained either silent or encouraged the dog to interact with the object, while an unfamiliar experimenter was present in the room in all conditions (Experiment 1). Moreover, we compared the dogs' behaviours in presence of either the owner or the stranger to when left alone in the room, with no additional experimenter present in the room (Experiment 2). We did so to understand if the sole presence of a stranger could improve dogs' performance in such problem-solving context (in comparison to when the dog is left alone), even if at a lesser extent than owners.

Based on the hypothesis that dogs form individualized attachment-like relationships specifically with their owner, we expected that dogs would: explore and manipulate the object for longer when the owner was with them than when a stranger was accompanying them, especially when the owner was encouraging them; be close to the owner for longer than to the stranger; spend more time close to the door and to the experimenter in presence of the stranger than when tested with the owner (Experiment 1 and 2). Nevertheless, following the hypothesis that strangers can also act as a secure base, we predicted that dogs would manipulate the object and explore for longer in the stranger presence than when left alone, and spend more time close to the door when left alone than in presence of the stranger (Experiment 2).

Concerning the differences between former shelter dogs and constant family dogs, we formulated three alternative hypotheses (and consequent predictions). (a) Dogs might be very flexible and could be able to build new bonds easily, even when adults [15, 23], resulting in no differences between constant family and former shelter dogs and a less strong owner-stranger differentiation. (b) Dogs' re-homing past might have an effect on their social development. This might lead to two alternative outcomes. (b1) Having lived in a shelter either negatively affects the future development of a secure attachment bond [26] or (b2) it increases the likelihood of accepting strangers as social partners [13, 15]. In the first case, former shelter dogs would manipulate and explore less in the presence of the owner than other family dogs. However, in the second case, the difference between former shelter dogs and other family dogs would appear in the time spent manipulating and exploring in the stranger condition. (c) Former shelter dogs might form a secure attachment bond with their owners, comparable to the one formed by other family dogs, but be generally more anxious than constant family dogs [6, 18]. Based on this, former shelter dogs would manipulate the apparatus and explore the environment less than constant family dogs with the stranger, while the two groups would not differ in the owner condition.

Overall, the present study allows us to test whether strangers could serve as a secure base for pet dogs and to compare the secure base effect of strangers to the one provided by owners. Moreover, the present study aims at investigating how having lived in a shelter may influence stranger acceptance and bonding with the owner.

## Materials and methods

### Ethical note

The experimental procedures were approved in accordance with Good Scientific Practice guidelines and national legislation by the Ethic Commission of the University of Veterinary Medicine Vienna (approval number: TK-02/10/2015). The owners were informed about the details of the procedures and gave their written consent to participate with their dogs.

### Experiment 1

**Subjects.** In the first experiment, we tested 56 adult dogs of different breeds (including mixed breeds) that were kept as pets in and near Vienna. Subjects were recruited from the

Clever Dog Lab database or via social media platforms. The dogs were at least one year old. Eight dogs had to be excluded due to failing the pre-test or because of technical issues with the video recording system. A total of 48 dogs (30 females and 18 males) completed the pre-test and the test (mean age ± SD = 4.17 ± 2.60 years, range: 1–10 years). Prior to the onset of the experiment, owners were informed that the aim of the study was to test the dog's attachment to its owner. In addition to the owner, three female experimenters participated in the study. One of them coordinated the experiment ("coordinating experimenter") whereas the other two took the roles of the two strangers participating in the test (see below).

Two groups of dogs were included: former shelter and constant family dogs. Based on the information provided by the owner, we included dogs into the group "Former Shelter Dogs" (n = 24) if they fulfilled the following criteria: 1) they had been at least three months old when they were brought into a shelter, 2) they had spent at least one month in the shelter, and 3) at the time of testing they had already been living with their owner for at least four months (following [14]). Dogs in the group "Family Dogs" (n = 24) were then selected to match the dogs in the "Former Shelter Dogs" group in respect to breed, sex and age. All constant family dogs had been living with the same owner since their age of 8–12 weeks, while dogs with a past in a shelter have been adopted either when they were less than one year old (26.09%), when they were between 1 and 4 years-old (43.48%) or older than 4 years old (30.43%).

**Experimental set-up.**   The experiments were carried out in an experimental room (3 m x 5 m) of the Clever Dog Lab (University of Veterinary Medicine Vienna, Austria). The room had two doors, one leading to the hallway and the other one leading to an adjacent room. The room was equipped with three cameras that were connected to a monitoring and recording station placed on the hallway. A chair, where the owner or the stranger was seated (depending on the condition, see below), was placed on the left side of the door leading to the hallway. Opposite to this door, a clock was hanging on the wall. The coordinating experimenter was present in the room standing on the right-hand side of the door and was watching the clock during the trials. At the beginning of each trial, she released the dog at a specific spot marked with tape on the floor. The apparatus (a folded kitchen towel in the case of the pre-test or one of four different toys baited with food in case of the test) was placed in the middle of the room (as shown in S1 Fig 1 in S1 File).

**Materials.**   The materials were the same folded kitchen towel and four customary dog toys as used in [11] (see S1 Fig 2 in S1 File). The apparatuses were filled with five pieces of preferred treats, e.g. sausages (∅ 1 cm). The dogs had to manipulate and move the toys either with their paw or muzzle to retrieve these five pieces of reward. We also added a bigger piece of sausage (about 3–5 cm long) that the dogs could not retrieve from the toys, in order to keep them motivated to further manipulate the toy after they had obtained all five treats.

**Experimental design and general procedure.**   Every dog had to pass a pre-test in order to proceed to the test. The pre-test aimed at ensuring that only dogs that were food-motivated enough to engage in manipulating an object also in absence of their owner would participate in the main test.

The pre-test as well as the test followed the same general procedure: first, the coordinating experimenter handled the food reward or baited the apparatus (the kitchen towel or one of the four dog toys) in the hallway in front of the dog. Afterwards she placed it on the marked spot in the experimental room while the dog was watching her from the hallway (the door to the experimental room stayed open) while held by the owner on a leash. Then the coordinating experimenter returned to the hallway and she took the dog's leash over from the owner and stayed with the dog in the hallway. The owner (in the "Owner" conditions) or the owner and the stranger (in the "Stranger" conditions) entered the experimental room and closed the door. The coordinating experimenter and the dog waited outside the testing room in the

hallway. The owner or the stranger sat down on the chair, depending on the condition. In the conditions where the stranger was present, the owner left the room through the door to the adjacent room and waited there until the trial was over. The owner could observe the test from the adjacent room, thanks to a monitoring system. The door between the experimental room and the adjacent room where the owner watched the test trial stayed closed throughout the trial.

The coordinating experimenter waited for 10 seconds after the owner had left, she opened the door, and then she entered the experimental room together with the dog. After closing the door behind them, she released the dog at the designated spot, and positioned herself next to the door. She remained there until the trial was over (until the dog had eaten all treats or after a maximum of one minute), continuously looking at the clock on the other side of the room. The coordinating experimenter did not interact with either the dog, or the owner, or the stranger.

At the end of the trial, the coordinating experimenter called the dog by its name and attached the leash to it. She left the testing room together with the dog, and closed the door. The two then waited in the hallway. In the owner conditions, the owner waited for ten seconds and then left the testing room using the door leading to the hallway. In the stranger conditions, the stranger waited until the coordinating experimenter closed the door, then picked the owner up from the adjacent room. They left the testing room together, using the door to the hallway. A break of approximately three minutes was carried out between trials when the dog was free to interact with its owner.

**Pre-test.**   The pre-test comprised four short trials repeated in a fixed order where the dogs needed to work more and more for the food, first in presence and then in absence of their owner.

Trial 1: five pieces of food were placed on the floor and the owner was present, seated on the chair next to the door and reading a magazine.

Trial 2: five pieces of food were placed on the floor, as in the first trial. However, after she/he had entered, the owner left the testing room through the door leading to the adjacent room where he or she waited until the trial was over.

Trial 3: the food was hidden in the folded cotton towel lying on the floor, and the owner was present like in the first trial.

Trial 4: the treats were again hidden in the folded towel as in the third trial but the owner was absent like in the second trial.

The dog passed the pre-test and could proceed to the test if she had eaten the treats at least once in the absence of its owner and in at least 3 out of 4 trials.

The coordinating experimenter was present in the testing room during the trials, thus the dog was never alone during the pre-test. Each trial ended when the dog had eaten all treats or after a maximum of one minute.

**Test.**   The test phase consisted of four different conditions. Each condition was conducted only once, therefore each dog participated in a total of four trials. The order of conditions was counterbalanced across dogs. Each trial lasted for five minutes. Within this period, the dog had the possibility to manipulate the apparatus in order to obtain the treats. In each trial, the dog received a different toy. The assignments of the four toys and the three experimenters to the different conditions were pseudo-randomized so that each toy and each experimenter were used evenly across conditions. As such, two different persons acted as strangers in the two stranger conditions.

Across conditions, we varied whether (in addition to the coordinating experimenter) the owner or the stranger (partner) was present in the room and how they behaved during the test: they either remained silent or encouraged the dog to manipulate the toy in order to retrieve a treat.

"Silent Owner" condition: the owner was present in the testing room but had to remain silent reading a magazine. The owner was not allowed to give commands, to point or to draw the dog's attention to the toy in any other way. However, the owner was allowed to pay attention to his/her dog and to pet it when it approached him/her.

"Encouraging Owner" condition: the owner was present and allowed to direct the dog's attention to the toy, thus motivating the dog verbally and through pointing gestures. Nevertheless, the owner had to remain seated throughout the trial.

"Silent Stranger" condition: a stranger sat on the owner's chair and behaved in the same way as the owner behaved in the "Silent Owner" condition.

"Encouraging Stranger" condition: a second stranger took the place of the owner and motivated the dog to manipulate the apparatus as described in the "Encouraging Owner" section.

**Behavioural analysis.** All experiments were videotaped and coded using Solomon Coder (© András Peter). In particular, the durations of the following behavioural variables were coded: "Manipulation" of the toy, "Close to partner" (either the owner or the stranger, depending on the testing condition), "Close to experimenter" (specifically referring to the coordinating experimenter), "Close to door", and "Exploration" of the room. All coded behaviours followed the definitions used in the study by [25]: "Manipulation" was coded whenever the dog was touching the toy with its muzzle or paw; "Close to partner" was recorded when the dog was in arm's reach of the respective person. If the dog was in reach of both persons present (coordinating experimenter and partner—owner or stranger), the dog was coded as being close to the person who was positioned closer to the head of the dog; "Close to door" was coded when the dog was standing, sitting or lying within the distance of one body length in front of one of the two doors and was looking at the door; "Exploration" was coded whenever the dog's muzzle was within 5 centimetres of any surface within the room. The videos were coded by two observers independently. Inter-observer reliability between the two of them was established by double coding 20% of the videos and calculating the Intra-class Correlation Coefficient (ICC). We obtained good or excellent levels of inter-observer reliability for all coded behaviours (ICC for Manipulation: 0.99; Close to Partner: 0.89; Close to Door: 0.99; Exploration: 0.77).

**Statistical analyses.** All behavioural variables were analysed as proportion of duration (amount of time in which the dog showed that specific behaviour divided per the total trial duration). Generalized Mixed Models (GLMM) with beta (Manipulation and Close to Experimenter) or beta inflated (Close to Door, Close to Partner, and Exploration) distribution were used to analyse whether the following predictors were associated with the behavioural variables analysed in the current study: partner (owner vs. stranger), behaviour of the partner (silent vs. encouraging), origin (constant family vs. former shelter), order of condition (1st, 2nd, 3rd or 4th), and the three two-way interactions between partner, behaviour of the partner, and origin. The sex of the dog was included in the model to control for it. The individual dog was included as random effect. Non-significant interactions were removed from the model. When the interaction between partner and origin was found to be significant, we investigated the effect of partner post-hoc separately for constant family and former shelter dogs (Bonferroni correction was applied to correct for multiple testing). Models were fitted using the package *gamlss* [27] in the software R (version: 3.6.2, R Core Team, 2019). Homoscedasticity and normality of model residuals were assessed via plots of residuals against fitted values and via qq-plots respectively. The threshold for significance was alpha = 0.05 for all tests.

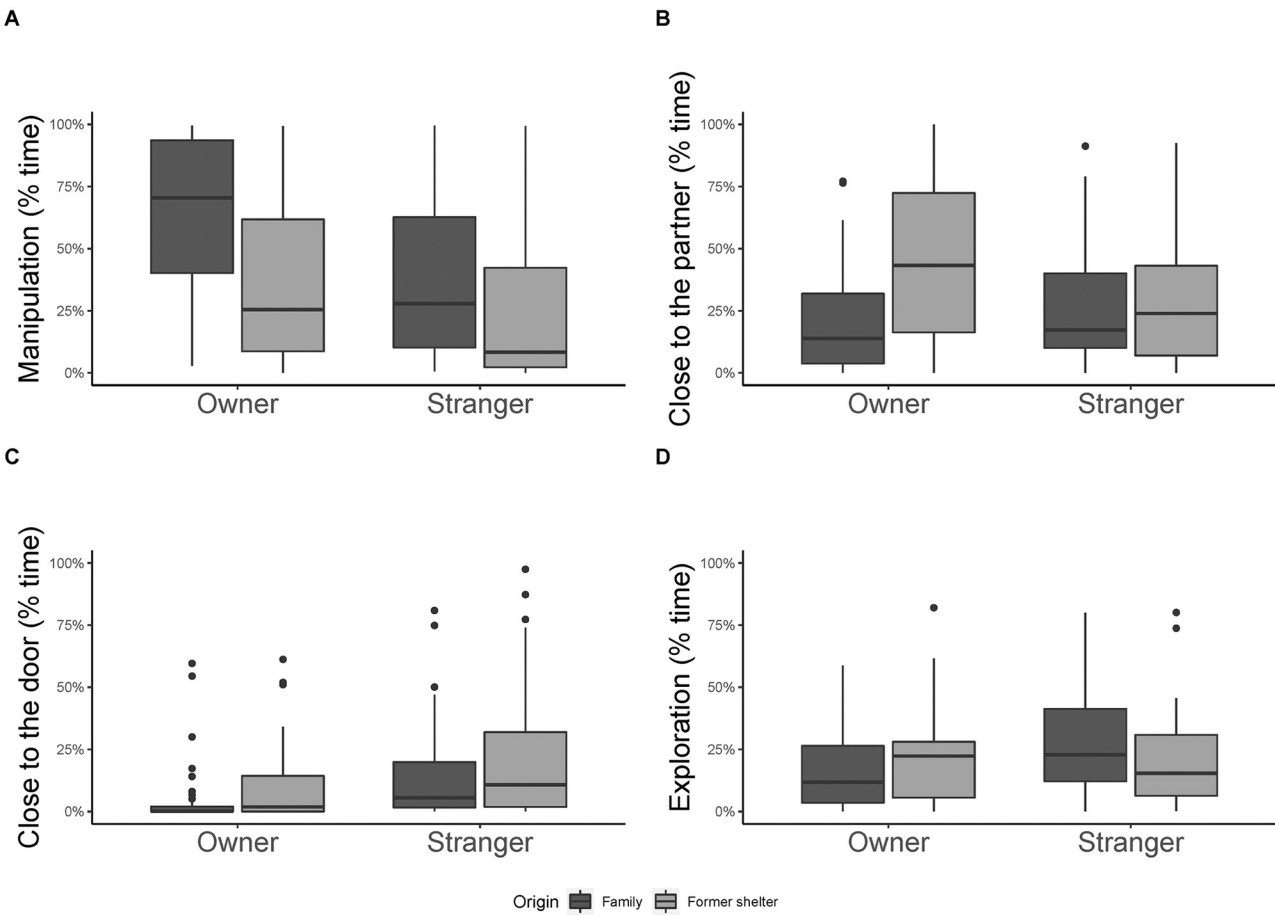

**Fig 1. Percentage of time spent showing the behaviours analysed in Experiment 1.** Percentage of time spent A) manipulating the object; B) close to the partner; C) close to the door; D) exploring in Experiment 1. Median and interquartile range (IQR; represented by the box), 25th percentile + 1.5 IQR, and 75th − 1.5 IQR (represented by the lower and the upper whiskers respectively).

**Results.** *Manipulation.* Dogs spent more time manipulating the apparatus in presence of the owner than in presence of the stranger (GLMM, estimate ± SE = 0.76 ± 0.15, df = 1, t = 5.03, p < 0.001, Fig 1), and constant family dogs spent more time manipulating the apparatus than former shelter dogs (GLMM, estimate ± SE = 1.04 ± 0.15, df = 1, t = 6.67, p < 0.001, Fig 1). Behaviour of the partner, order of condition and all interactions resulted not significant (all p > 0.05, S1 Table 1 in S1 File).

*Close to the partner.* Dogs spent more time close to the partner when this was encouraging them than when she/he was silent (GLMM, estimate ± SE = 0.35 ± 0.14, df = 1, t = 2.63, p = 0.01). In addition, we found a significant effect of the interaction between the partner and the origin of the dog (GLMM, estimate ± SE = 0.94 ± 0.27, df = 1, t = 3.47, p < 0.001, Fig 1). In particular, former shelter dogs spent more time close to the owner than to the stranger (GLMM, estimate ± SE = 0.59 ± 0.20, df = 1, t = 2.96, p = 0.004, Fig 1), while constant family dogs spent equal time with both partners (GLMM, estimate ± SE = 0.30 ± 0.19, df = 1, t = 1.52, p = 0.13, Fig 1). Overall dogs spent more time close to the partner as sessions went by (GLMM, estimate ± SE = 0.14 ± 0.06, df = 1, t = 2.32, p = 0.02). The other interactions resulted not significant (all p > 0.05, S1 Table 1 in S1 File).

*Close to the door*. Dogs spent more time close to the door when the stranger was present than when the owner was present (GLMM, estimate ± SE = 0.69 ± 0.16, df = 1, t = 4.18, p < 0.001, Fig 1), more in the silent trials than in the encouraging trials (GLMM, estimate ± SE = 0.52 ± 0.16, df = 1, t = 3.33, p = 0.001) and former shelter dogs did so longer than constant family dogs (GLMM, estimate ± SE = 0.56 ± 0.16, df = 1, t = 3.54, p < 0.001, Fig 1). None of the interactions affected the time spent close to the door (all p > 0.05, S1 Table 1 in S1 File).

*Exploration*. Dogs spent more time exploring when the partner was encouraging them than when she/he was silent (GLMM, estimate ± SE = 0.37 ± 0.11, df = 1, t = 3.43, p < 0.001). In addition, we found a significant interaction between partner and origin of the dog (GLMM, estimate ± SE = 0.67 ± 0.22, df = 1, t = 3.05, p = 0.003). In particular, constant family dogs spent more time exploring when tested with the stranger than when tested with the owner (GLMM, estimate ± SE = 0.68 ± 0.16, df = 1, t = 4.19, p < 0.001, Fig 1), while no such difference was found in shelter dogs (GLMM, estimate ± SE = -0.02 ± 0.16, df = 1, t = -0.13, p = 0.89, Fig 1). Overall dogs spent less time exploring as sessions went by (GLMM, estimate ± SE = -0.11 ± 0.05, df = 1, t = -2.17, p = 0.03). All other interactions resulted not significant (all p > 0.05, S1 Table 1 in S1 File).

*Close to the experimenter*. Dogs spent more time close to the coordinating experimenter in the stranger's presence than when the owner was present (GLMM, estimate ± SE = 0.61 ± 0.15, df = 1, t = 3.96, p < 0.001, Fig 2) and more if they were constant family dogs than if they were former shelter dogs (GLMM, estimate ± SE = 0.63 ± 0.15, df = 1, t = 4.13, p < 0.001, Fig 2). Overall dogs spent less time close to the coordinating experimenter as sessions went by (GLMM, estimate ± SE = -0.19 ± 0.07, df = 1, t = -2.85, p = 0.005). Behaviour of the partner and all interactions resulted not significant (all p > 0.05, S1 Table 1 in S1 File).

## Experiment 2

**Subjects.**  We recruited 30 adult dogs following the same selection process and criteria as in the first experiment. The mean age ± SD was 5 ± 3.71 years, range: 1–15 years. Six dogs had to be excluded due to failing the pre-test or because of technical issues with the monitoring system. A total of 24 dogs (15 females and 9 males) passed the pre-test and completed the test (mean age ± SD = 4.79 ± 3.46 years, range: 1–15 years). Twelve dogs were "Former shelter dogs" while the other 12 were "Family dogs". Dogs were selected and assigned to the two groups using the same criteria as for Experiment 1. Also in this case all constant family dogs have been acquired when younger than 12 months, while former shelter dogs were adopted when younger than one year (9.09%), between one and 4 years of age (63.64%) and when older than 4 years (27.27%).

Differently from the previous experiment, the coordinating experimenter was not present in the testing room during testing, instead she observed the experiment through the monitoring system from the hallway. Only two experimenters participated in this experiment since the number of trials was reduced (i.e. only one stranger condition, see below). Their assignment as stranger or coordinating experimenter to the single tests was again pseudo-randomized.

**Materials and experimental set-up.**  The experimental room remained the same as in Experiment 1, except that the clock had been removed since the coordinating experimenter was not present in the testing room in Experiment 2. Furthermore, we excluded the blue wobbling dog toy (see S1 Fig 2 in S1 File) which had been used in the first experiment, since the number of trials was reduced to three.

**Experimental design and general procedure.**  The general procedure and the experimental design were similar to those in Experiment 1, except that the coordinating experimenter

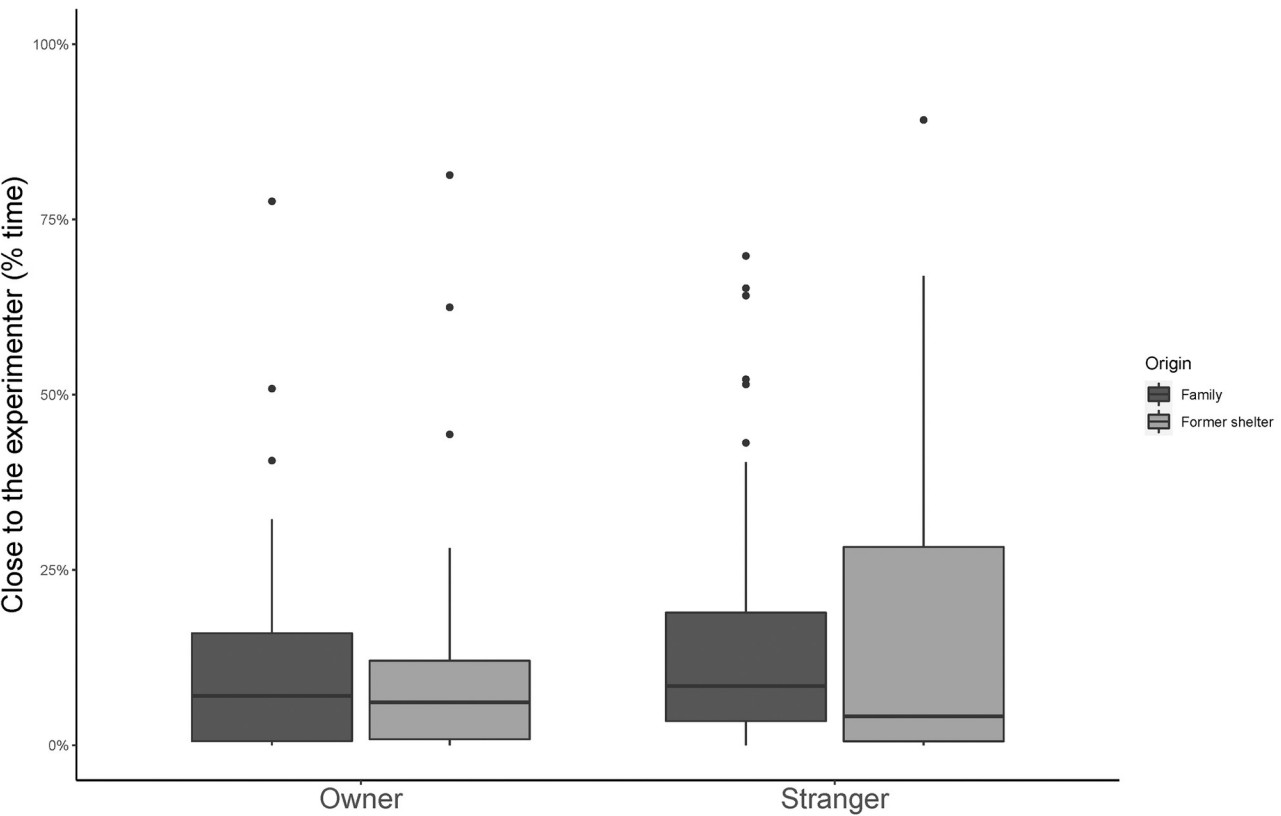

**Fig 2. Percentage of time spent close to the coordinating experimenter in Experiment 1.** Median and interquartile range (IQR; represented by the box), 25th percentile + 1.5 IQR, and 75th − 1.5 IQR (represented by the lower and the upper whiskers respectively).

was not present anymore in the testing room during the trials. For this aim, the coordinating experimenter returned to the hallway after unleashing the dog at the release point. During the trials, she waited outside of the room, and observed the trials through the monitoring equipment. When the trial was over, the coordinating experimenter entered the room again, called the dog by its name and leashed it before leaving the room.

**Pre-test.**   Each dog participated in the same four pre-test trials as in Experiment 1, with the only exception that the coordinating experimenter was not in the room during the trials.

**Test.**   We conducted three trials in this experiment, as follows. As in the first study, the order of the trials was counterbalanced across dogs and the assignment of the three toys was pseudo-randomized.

"Owner" condition: followed the procedure of the "Silent Owner" condition of Experiment 1, with the exception that the coordinating experimenter was not present in the room.

"Stranger" condition: followed the procedure of the "Silent Stranger" condition of Experiment 1, with the exception that the coordinating experimenter was not present in the room.

"Alone" condition: the owner waited in the hallway during the whole trial. The coordinating experimenter released the dog in the experimental room and immediately went back to the hallway, living the dog alone in the experimental room.

**Behavioural analysis.**   The same behaviours as in Experiment 1 were coded. However, in the "Alone" condition, "Close to partner" was coded when the dog was within the distance of one body length to the empty chair.

**Statistical analyses.** All behavioural variables were analysed as proportion of duration (amount of time in which the dog showed that specific behaviour divided per the total trial duration). Generalized Mixed Models (GLMM) with beta (Close to Partner and Exploration) or beta inflated (Manipulation and Close to Door) distribution were used to analyse whether the following predictors were associated with the behavioural variables analysed in the current study: partner (owner vs. stranger vs. alone), origin (constant family vs. former shelter), order of condition (1st, 2nd, or 3rd), and the two-way interaction between partner and origin. The sex of the dog was included in the model to control for it. The individual dog was included as random effect. Whenever the two-way interaction was non-significant, it was removed from the model. When the interaction between partner and origin was found to be significant, we investigated the effect of partner post-hoc separately for constant family and former shelter dogs (Bonferroni correction was applied to correct for multiple testing). Models were fitted using the package *gamlss* [27] in the software R (version: 3.6.2, R Core Team, 2019). Homoscedasticity and normality of models' residuals were assessed via plots of residuals against fitted values and via qq-plots. The threshold for significance was alpha = 0.05 for all tests.

**Results.** *Manipulation.* Dogs spent more time manipulating the apparatus in presence of the owner than in presence of the stranger (GLMM, estimate ± SE = 0.86 ± 0.28, df = 1, t = 3.03, p = 0.004, Fig 3) and more with their owner than when they were tested alone

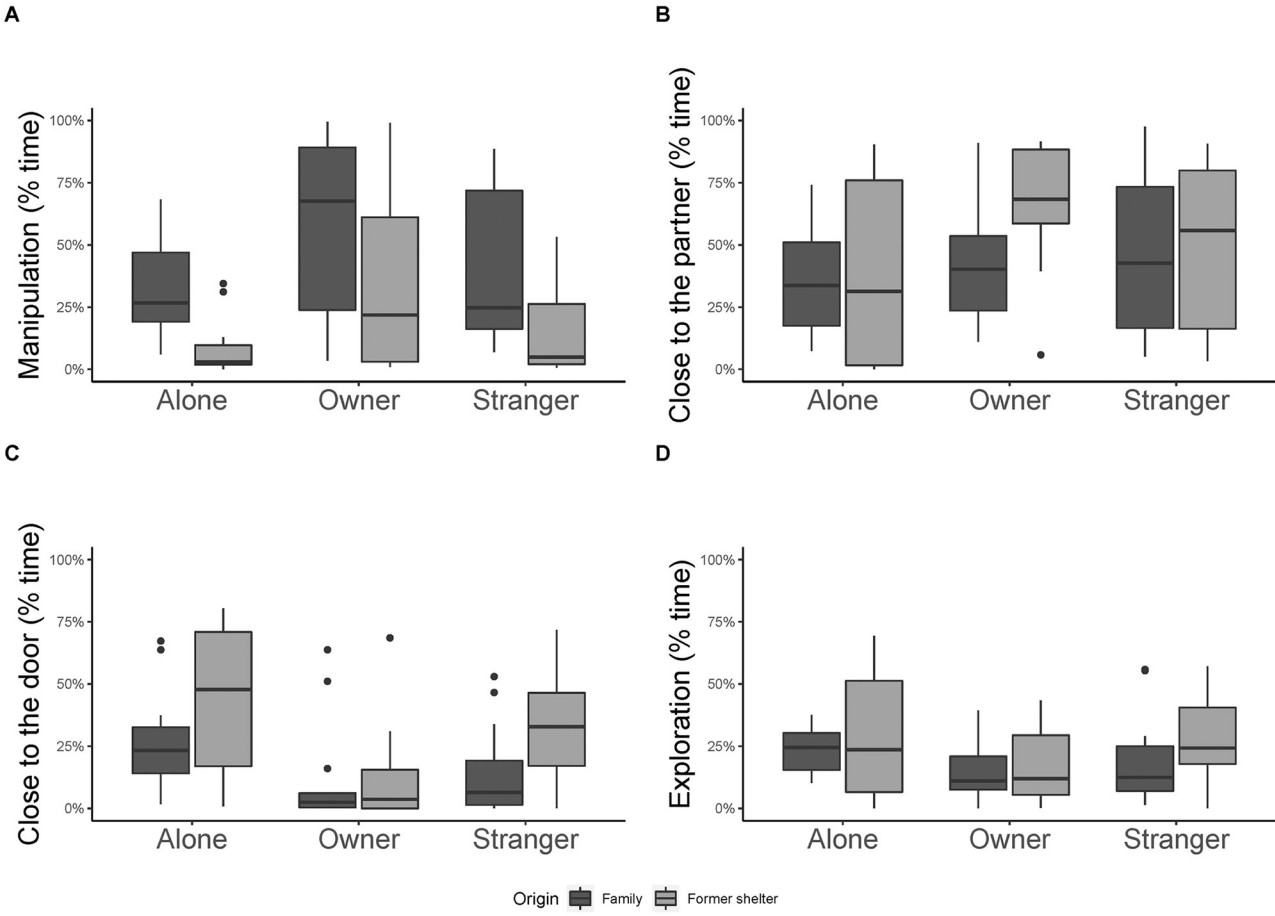

**Fig 3. Percentage of time spent showing the behaviours analysed in Experiment 2.** Percentage of time spent A) manipulating the object; B) close to the partner; C) close to the door; D) exploring in Experiment 2. Median and interquartile range (IQR; represented by the box), 25th percentile + 1.5 IQR, and 75th − 1.5 IQR (represented by the lower and the upper whiskers respectively).

(GLMM, estimate ± SE = 1.09 ± 0.30, df = 1, t = 3.64, p < 0.001, Fig 3). No difference was found between the condition in which dogs were tested alone or with the stranger (GLMM, estimate ± SE = 0.22 ± 0.30, df = 1, t = 0.73, p = 0.47, Fig 3). Moreover, constant family dogs spent more time manipulating the apparatus than former shelter dogs (GLMM, estimate ± SE = 1.20 ± 0.25, df = 1, t = 4.74, p < 0.001, Fig 3). The order of condition and inter-action between partner and origin of the dog resulted not significant (all p > 0.05, S1 Table 2 in S1 File).

*Close to the partner*. We found a significant effect of the interaction between the partner and the origin of the dog. In particular, shelter dogs spent more time close to the owner than close to the chair in the alone condition (GLMM, estimate ± SE = 1.32 ± 0.44, df = 1, t = 2.98, p = 0.006, Fig 3) but there was no difference between the stranger and owner condition or between the stranger and the alone condition (both p > 0.05). There were also no significant differences across conditions in constant family dogs (all p > 0.05, S1 Table 2 in S1 File). The order of condition resulted not significant (p > 0.05, S1 Table 2 in S1 File).

*Close to the door*. Dogs spent more time close to the door when tested alone than when tested with the owner (GLMM, estimate ± SE = 1.05 ± 0.30, df = 1, t = 3.51, p < 0.001, Fig 3). However, no difference was found between the stranger and the owner condition (GLMM, estimate ± SE = 0.65 ± 0.31, df = 1, t = 2.13, p = 0.04, not significant after correction for multi-ple comparisons, Fig 3), nor between the stranger and the alone condition (GLMM, estimate ± SE = 0.40 ± 0.26, df = 1, t = 1.50, p = 0.14, Fig 3). Shelter dogs spent more time close to the door than constant family dogs (GLMM, estimate ± SE = 0.70 ± 0.24, df = 1, t = 2.91, p < 0.001, Fig 3). The order of condition and interaction between partner and origin of the dog resulted not significant (all p > 0.05, S1 Table 2 in S1 File).

*Exploration*. Dogs spent less time exploring when tested with the owner than when tested alone (GLMM, estimate ± SE = -0.64 ± 0.25, df = 1, t = -2.61, p = 0.01, Fig 3). The proportion of time dogs spent exploring did not differ when they were tested with the stranger or alone (both p > 0.05, Fig 3, S1 Table 2 in S1 File). The order of condition, the origin of the dog, and the interaction between partner and origin of the dog resulted not significant (all p > 0.05, S1 Table 2 in S1 File).

## Discussion

The present study had three main aims: investigate whether strangers could serve as a secure base for pet dogs and to what extent they would do so less than owners. Thirdly, as an owner-stranger differentiation might depend on a dog's life history, we aimed to examine whether the secure base effect provided by owners (and potentially by strangers) would differ between dogs with a past in the shelter and dogs who have always lived with the same owner. We did so by conducting two experiments where dogs faced a problem-solving task in the presence of the owner or a stranger. We varied the behaviour of the owner and the stranger (either remaining silent or encouraging the dog, Experiment 1), the presence of an unfamiliar coordinating experimenter in the room (only present in Experiment 1), and the overall presence of a partner (i.e. the dog was alone in the room in one condition of Experiment 2).

One of our main findings is that none of the two experiments found evidence that strangers could provide dogs with a secure base effect. In fact, no difference between the alone and the stranger conditions could be found in Experiment 2. However, we could confirm previous evidence that owners' presence is a source of security for dogs [3, 25, 28]. We found this effect to be independent from the owner's behaviour (either remaining silent or encouraging the dog) and consistent across both experiments. As such, we found that dogs, independent of their past, manipulated the object longer in presence of the owner than in presence of the stranger

(both in the Experiment 1 and 2) or when left alone (Experiment 2). Furthermore, dogs spent more time close to the door (both in Experiment 1 and 2) or to the coordinating experimenter (Experiment 1) when they were alone or when the stranger was present than when the owner was present, suggesting that dogs were less comfortable in the experimental environment when separated from their owners than when the owner was in the room. These results confirm that dogs form individualized relationships with their owner [3, 25, 28] and do not seem to use a stranger as a secure base in this problem-solving context. Importantly, results of the two experiments were consistent, suggesting that the presence of an unfamiliar coordinating experimenter in Experiment 1 (in addition to the person acting as a partner) did not play a major role in affecting the behaviour of the subjects.

It is important to note that in this study the person acting as a stranger had never interacted with the dog before the onset of the experiment. It is possible that these results would have been different if a short familiarization phase had preceded the test. In fact, previous studies have shown that dogs might show attachment-like behaviours towards a human even if this has interacted with them only three times for a few minutes [15]. Future studies will need to address whether and when dogs can benefit from the presence of other humans than their owners, depending on the degree of their familiarity (e.g. after one or more interaction sessions).

Differently from Horn et al.'s study using a similar procedure [25], here we did not find that encouragement had an effect on dogs' object manipulation. However, we found that dogs explored more, spent more time close to the partner, and less time close to the door when the partner was giving encouragements than when he/she was silent, suggesting that generally dogs were more active in the encouraging than in the silent conditions. This finding highlights that similar studies addressing the effect of human communication on dogs' problem solving need to control for the owners' effect on the general activity of the dogs that may indirectly increase (or in certain cases decrease, see [29]) the problem solving success of the dogs.

Concerning our second question (whether owner-stranger differentiation depends on the dogs' life history), we found that the above-mentioned general effects did not depend on the origin of the dogs. In fact, the two tested groups (former shelter and constant family dogs) differed neither in terms of owner-alone differentiation in object manipulation, proximity to the door, room exploration nor in owner-stranger differentiation in terms of object manipulation and proximity to the door. Furthermore, encouragement affected staying in proximity to door and to the partner, and exploration similarly in both groups of dogs. Nevertheless, we did find other differences between the former shelter and constant family dogs. First, in Experiment 1 (but not in Experiment 2) we found that exploration was not affected by the identity of the partner in former shelter dogs, while constant family dogs explored the environment longer in the presence of the stranger than in the presence of the owner. This latter pattern observed in constant family dogs seems to suggest that exploration may have been more motivated by the animals' distress (evoked by the owner's absence or the stranger's presence) rather than by their desire to learn about the environment (that should have rather been increased when the owner's presence provided a secure base). If exploration was a sign of distress also in the case of former shelter dogs, then a similar duration of exploration in the presence of the owner and of the stranger might be either due to low stress when the owner is absent (stranger condition) or to received support when the owner is present (owner condition). As former shelter dogs spent more time close to the owner than to the stranger (both in Experiment 1 and 2), it is possible to assume that former shelter dogs could use their owners as a secure base to a lesser extent than constant family dogs. However, as former shelter dogs manipulated the objects more and spent less time close to the door in presence of their owner than in presence of the

stranger to an extent similar to constant family dogs, we have no clear evidence that these two groups of dogs would have different bonds to their owners.

Rather, we suggest that the differences between former shelter and constant family dogs described above, including less object manipulation and more time spent close to the door (both in Experiment 1 and 2) in former shelter dogs, show that former shelter dogs were less confident with interacting with the novel object and with being in the experimental room than constant family dogs. Importantly, former shelter dogs' general wariness did not seem to be only related to the physical environment but also to the stranger. In fact, former shelter dogs spent less time close to the coordinating experimenter than constant family dogs (Experiment 1) and they spent less time close to the stranger than to their owners (while constant family dogs did not show such a difference, Experiment 1). These results are in line with the only previous study comparing former shelter dogs to other family dogs in the SST, also suggesting that former shelter dogs are able to form an attachment bond with their owner but also that they might be more anxious when facing novel stimuli than other family dogs [6, 17]. However, although these results are in line with previous evidence showing that shelter dogs are more fearful when confronted with an unfamiliar human than pet dogs [30, 31], we recommend caution in comparing the behaviour of dogs when still housed in a stressful environment like the shelter [19–21] to the one of dogs who have been living as pets already since months (as in the present study). Taken together, the differences between the two groups do not seem to depend on the relationship with the owner per se, but rather on former shelter dogs' perceiving the test situation more stressful (as in [6, 17]).

What factors are the exact cause of the above-mentioned differences between former shelter dogs and constant family dogs remains to be explored. In fact, multiple elements might have contributed to such differences: lower manipulation time could be due to having had fewer chances to interact with novel human artefacts making former shelter dogs more neophobic [32] or less persistent [33] than constant family dogs. Traumatic experiences connected to the relinquishment or their stay in the shelter may have led to the development of learned helplessness [34, 35], hindering their motivation to engage in problem-solving activities. Moreover, differences in the interaction with the social partner (i.e. shelter dogs spending less time close to the coordinating experimenter and more time close to the owner than to the stranger in comparison to constant family dogs) could be due to potential previous mistreatment [36, 37], traumatic experience before the shelter [36], and/or in the shelter [19–21], and potentially to the disruption of the first attachment bond to their first owners [38]. Additionally, it cannot be excluded that the dogs in our former shelter dog group had differed from members of the constant family dog group already before ending up in the shelter because of potential behavioural problems (e.g. anxiety) that might have emerged before the relinquishment (and potentially be the reason for it, [22]). Or that owners adopting from shelters might be different to those adopting puppies from private hands, with adopters of shelter dogs potentially being more aware and more willing to accept behavioural problems (e.g. anxiety) than owners acquiring a dog from a breeder [39, 40]. Being more tolerant towards potential behavioural problems might influence how an owner interacts with his/her dog, ultimately influencing a dog's behaviour in novel situations [12]. For instance, more tolerant owners might be less inclined to punish their dogs when these show a problematic behaviour [41] or show a warmer and more affectionate interaction style, increasing their dogs' reliance on their guidance and their sense of security [12]. However, to the best of our knowledge, no study has investigated whether adopters from shelters behave differently towards their dogs than other dog owners and whether these differences influence their dogs' behaviour yet.

An additional element that future studies would need to take into account is the time that adopted dogs have spent with their new owner before the onset of the experiment. A previous

study, in fact, have shown that dogs' attachment towards their owners depended on how long they have been adopted [42]. The present sample size was not big enough to investigate the effect of this variable, which might have contributed to the within-group variability in former shelter dogs. We recommend future studies to take this element into account, as it might affect the ease with which former shelter dogs accept strangers as well as their emotional reactivity to novel stimuli [42].

Our findings have important practical implications. Despite some general differences to constant family dogs, former shelter dogs also proved to be able to benefit from their owner's secure base effect, suggesting that adopting an adult dog from a shelter does not hinder the development of an attachment bond. Showing that re-homed dogs can form attachment bonds to their human caregivers comparable to bonds of dogs that have always lived in the same family could improve humans' attitudes to adoption from shelters (potentially increasing the adoption rate, [40]). Still, differences related to motivation in object manipulation and anxiety in a novel situation have emerged, suggesting that previous experiences might affect dogs' coping with novel environments and problem solving performance. Being aware of such behavioural characteristics can help owners have more realistic expectations of their dogs, reducing the risk of frustration and potential dissatisfaction, both potentially leading to a weak dog-human bond and relinquishment. Understanding how owners can provide security to their dogs when interacting with the environment is utterly important for those dogs who come from a history of deprivation and who need to be introduced to the complexities of the human environment. Understanding whether former shelter dogs differ from dogs adopted in puppyhood can help not only adopters, but also help design specific interventions to enrich the lives of dogs hosted in shelters, to support owners in their interactions with their newly adopted dogs, and to familiarize dogs with environments they might live in after adoption.

## Supporting information

**S1 File. Sketch of the experimental room, materials used, tables with complete models results from Experiment 1 and 2.**
(DOCX)

**S2 File. Complete dataset.**
(XLSX)

## Acknowledgments

We are most thankful to Lisa Horn for contributing to the initial plan of this study, to Sarah Marshall-Pescini and all IMHAI teachers for discussing it at intermediate stages, and to Lisa Wallis for taking part of the supervision. We thank Karin Bayer for administrative support, and Wolfgang Berger and Peter Füreder for technical support. In addition, we sincerely thank all owners and dogs for their participation in the study.

## Author Contributions

**Conceptualization:** Julia Schindlbauer, Teresa Pegger, Verena Wesian, Zsófia Virányi.

**Data curation:** Giulia Cimarelli, Julia Schindlbauer, Teresa Pegger, Verena Wesian.

**Formal analysis:** Giulia Cimarelli.

**Investigation:** Julia Schindlbauer, Teresa Pegger, Verena Wesian.

**Methodology:** Julia Schindlbauer, Teresa Pegger, Verena Wesian.

**Project administration:** Giulia Cimarelli, Zsófia Virányi.

**Resources:** Zsófia Virányi.

**Supervision:** Giulia Cimarelli, Zsófia Virányi.

**Visualization:** Giulia Cimarelli.

**Writing – original draft:** Giulia Cimarelli.

**Writing – review & editing:** Giulia Cimarelli, Zsófia Virányi.

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
