## [Decision Letter · Decision Letter 0]

8 Sep 2021

PONE-D-21-22971Secure base effect in former shelter dogs and other family dogs: strangers do no provide security in a problem-solving taskPLOS ONE

Dear Dr. Cimarelli,

Thank you for submitting your manuscript to PLOS ONE. After careful consideration, we feel that it has merit but does not fully meet PLOS ONE’s publication criteria as it currently stands. Therefore, we invite you to submit a revised version of the manuscript that addresses the points raised during the review process. For details, see the review report below.

We look forward to receiving your revised manuscript.

Kind regards,

I Anna S Olsson, Ph.D.

Academic Editor

PLOS ONE

Journal Requirements:

Reviewers' comments:

Reviewer's Responses to Questions

**Comments to the Author**

1. Is the manuscript technically sound, and do the data support the conclusions?

Reviewer #1: Yes

2. Has the statistical analysis been performed appropriately and rigorously? 

Reviewer #1: Yes

3. Have the authors made all data underlying the findings in their manuscript fully available?

Reviewer #1: Yes

4. Is the manuscript presented in an intelligible fashion and written in standard English?

Reviewer #1: Yes

5. Review Comments to the Author

Reviewer #1: The present manuscript reports a study on dog-owner attachment. The goal of the study was twofold: 1) to disentangle between the secure-base effect of owner vs stranger and 2) to investigate how a dog’s history (namely, having lived in the same household since puppyhood vs having lived part of life in a shelter) would influence this secure-base effect. To that end, in two experiments, shelter and family dogs were exposed to a problem-solving task where it was varied whom was present with the dog in the experimental room (owner, stranger, alone), as well as the behavior of the partner (encouraging, silent). It was hypothesized that, if only owners provide a secure-base effect to their dogs, dogs would spend more time trying to solve the problem when the owner was present than when the stranger was present or the dog was alone. Secondly, encouragement was expected to facilitate object manipulation (i.e., dogs would persevere more), at least in the presence of the owner. Finally, differences in behavior were expected to emerge as a function of dogs’ past history. Three major results were found:

1) Owners, but not strangers, provided dogs with a secure base effect – dogs manipulated the object longer in the presence of the owner than in the presence of the stranger or when alone;

2) This effect was independent of the owner’s behavior (silent vs encouraging)

3) The above mentioned effects did not depend on dogs’ past history. However, there were some differences in the behavior of the two groups, with shelter dogs being overall warier in the novel environment and in relation to the stranger (exhibiting more behavioral signs correlated with anxiety).

Authors concluded that both dogs that have lived since puppyhood in the same household and dogs that have been re-homed can develop a secure attachment to their owners, but that having spent some time in a shelter can hinder these dogs’ ability to cope with new environments and people.

The study was conducted with rigor, the methods are overall well-presented and the results support the conclusions. However, some more effort has to be put in the Introduction section, on the presentation of the state of the art as well as on the derivation and presentation of hypothesis. Specifically, because the study has two research questions that are answered by crossing/combining the results of the two conducted experiments, the authors need to further work on presenting the state of the art that informs their two questions and to better explain how they relate to each other, as well as to clarify the derivation of their hypothesis. I think this can be achieved mainly with some re-writing and re-organization of the Introduction section. I detail my comments in what follows.

Introduction

First, I think the Introduction needs to be shortened. I feel five pages is a lot concerning the size of the current paper but, more importantly, shortening will allow the authors to be more straightforward and direct when presenting the theoretical background and the gap that the current piece of research aims at filling. Some examples/suggestions are:

- Condense what is presented in paragraphs between lines 114 and 142. I think such a detailed presentation of Horn’s study is not necessary.

- Paragraphs between line 143 and 171: Can the authors condense and present the ideas in a more straightforward way? The level of methodological detail given here is, in my opinion, not necessary.

- As it is currently written, it almost seems that the Introduction will end on line 113. However, after that, the authors still discuss the issue on disentangling between owner and stranger secure-base effects, and afterwards they go back to shelter vs family dogs again. I would advise a reorganization in a way where the reader is not jumping from one topic to the other. An additional suggestion is that the hypotheses are only presented in the final paragraph of the Introduction. Personally, I do not think the authors need to go into so much detail regarding hypotheses in this section (currently there are several hypotheses presented across the Introduction). I would rather opt by presenting more general hypotheses (and the specific goals of each Experiment, which is not done in the current version of the manuscript) and maybe latter (for example, in the Methods section) present all the more “specific” hypotheses in detail. If the authors decide they want to present all the detailed hypotheses, I would strongly advise presenting them using bullet points, such that it is easier for the reader to follow and consult when needed.

Another comment I have to the Introduction concerns the ideas presented in lines 59-64:

“It is important to realise, however, that such an owner-stranger differentiation depends not only on the strength of the dog’s bond to its owners but also on the ease with which a dog accepts strangers as social partners. Stranger acceptance varies greatly across individual dogs and does so independent of the dogs’ bonds to their owners [3,10–12]. Many studies, however, do not make an effort to disentangle these two factors…” -- I wonder whether the authors aimed at disentangling this owner bond vs stranger acceptance effects in the present study. If they did, it is not currently clear how. If they did not, then maybe this specific issue can be dropped out of the Introduction.

Line 162-163: Please state earlier that an experimenter was always present in the room in Experiment 1, such that it is clear from early on that this was the case.

Methods

Please state early on in this section (in the Experimental Design and General Procedure) when the experimenter was or was not present in the room (in Experiment 1, Experiment 2 and pre-tests).

Line 245 - Who was holding the dog while “she placed it on the marked spot in the experimental room”?

Lines 248-249 - Why this difference between conditions? Why didn’t the owner and the stranger entered together as well in the “owner condition”?

Line 259: “At the end of the trial” – The information on what determined the end of a trial should be presented already here (currently it is only presented later, on line 270)

Line 260 – Please add to where the experimenter “left the testing room together with the dog” (hallway?)

The info on lines 271-280 should come in line 268 right after “first in presence and then in absence of their owner”.

Lines 406-408: I think the paper benefits if this info is presented earlier (see comment above).

Results

I think there are too many graphs/tables in this section. Please think reducing the number of figures – maybe create two big figures (one for Experiment 1 and one for Experiment 2) with several panels. Moreover, I would suggest moving the tables to supplementary material.

Discussion

My only comment to this section is that I missed seeing the present results compared/contrasted with those from Previde and Valsecchi (2007) [6, 17], the only other published study comparing attachment in shelter vs ‘family’ dogs.

6. PLOS authors have the option to publish the peer review history of their article (what does this mean?). If published, this will include your full peer review and any attached files.

Reviewer #1: **Yes: **Ana Catarina Vieira de Castro

---

## [Author Response · Author response to Decision Letter 0]

8 Oct 2021

Rebuttal letter

Manuscript PONE-D-21-22971 “Secure base effect in former shelter dogs and other family dogs: strangers do no provide security in a problem-solving task”

Dear Editor,

Thank you for considering the present manuscript for publication in PLOS ONE. We are pleased to see that the manuscript was positively received and that only few points have been raised by the reviewers that would need to be revised.

Please find below the answers to the points raised by you and by the reviewers. The answers are reported in bold immediately after each point.

Kind regards,

Giulia Cimarelli, PhD (corresponding author)

Done.

2. Please review your reference list to ensure that it is complete and correct. 

Done.

Review Comments to the Author

Reviewer #1: The present manuscript reports a study on dog-owner attachment. The goal of the study was twofold: 1) to disentangle between the secure-base effect of owner vs stranger and 2) to investigate how a dog’s history (namely, having lived in the same household since puppyhood vs having lived part of life in a shelter) would influence this secure-base effect. To that end, in two experiments, shelter and family dogs were exposed to a problem-solving task where it was varied whom was present with the dog in the experimental room (owner, stranger, alone), as well as the behavior of the partner (encouraging, silent). It was hypothesized that, if only owners provide a secure-base effect to their dogs, dogs would spend more time trying to solve the problem when the owner was present than when the stranger was present or the dog was alone. Secondly, encouragement was expected to facilitate object manipulation (i.e., dogs would persevere more), at least in the presence of the owner. Finally, differences in behavior were expected to emerge as a function of dogs’ past history. Three major results were found:

1) Owners, but not strangers, provided dogs with a secure base effect – dogs manipulated the object longer in the presence of the owner than in the presence of the stranger or when alone;

2) This effect was independent of the owner’s behavior (silent vs encouraging)

3) The above mentioned effects did not depend on dogs’ past history. However, there were some differences in the behavior of the two groups, with shelter dogs being overall warier in the novel environment and in relation to the stranger (exhibiting more behavioral signs correlated with anxiety).

Authors concluded that both dogs that have lived since puppyhood in the same household and dogs that have been re-homed can develop a secure attachment to their owners, but that having spent some time in a shelter can hinder these dogs’ ability to cope with new environments and people.

The study was conducted with rigor, the methods are overall well-presented and the results support the conclusions. However, some more effort has to be put in the Introduction section, on the presentation of the state of the art as well as on the derivation and presentation of hypothesis. Specifically, because the study has two research questions that are answered by crossing/combining the results of the two conducted experiments, the authors need to further work on presenting the state of the art that informs their two questions and to better explain how they relate to each other, as well as to clarify the derivation of their hypothesis. I think this can be achieved mainly with some re-writing and re-organization of the Introduction section. I detail my comments in what follows.

Introduction

First, I think the Introduction needs to be shortened. I feel five pages is a lot concerning the size of the current paper but, more importantly, shortening will allow the authors to be more straightforward and direct when presenting the theoretical background and the gap that the current piece of research aims at filling. Some examples/suggestions are:

We shortened and restructured the Introduction, making it more concise than before.

- Condense what is presented in paragraphs between lines 114 and 142. I think such a detailed presentation of Horn’s study is not necessary.

We drastically reduced this part and condensed the previous three paragraphs into one (lines 115-131).

- Paragraphs between line 143 and 171: Can the authors condense and present the ideas in a more straightforward way? The level of methodological detail given here is, in my opinion, not necessary.

We reduced the explanation of the methods to the minimum. The paragraph now extends from line 132 to line 139.

- As it is currently written, it almost seems that the Introduction will end on line 113. However, after that, the authors still discuss the issue on disentangling between owner and stranger secure-base effects, and afterwards they go back to shelter vs family dogs again. I would advise a reorganization in a way where the reader is not jumping from one topic to the other. 

We restructured the Introduction according to the following: general intro about the secure base effect � the secure base effect in dogs and the problem with previous studies: owner-stranger differentiation not always clear, especially for those dogs changing caregiver throughout their lifetime (e.g. guide dogs) � presentation of the phenomenon that former shelter dogs might differ from family dogs acquired since puppyhood in owner-stranger differentiation because of their life experiences � short presentation of the methods and aim for each experiment � hypotheses and predictions for the overall owner-stranger differentiation and for the former shelter vs. constant family dogs potential differences.

An additional suggestion is that the hypotheses are only presented in the final paragraph of the Introduction. Personally, I do not think the authors need to go into so much detail regarding hypotheses in this section (currently there are several hypotheses presented across the Introduction). I would rather opt by presenting more general hypotheses (and the specific goals of each Experiment, which is not done in the current version of the manuscript) and maybe latter (for example, in the Methods section) present all the more “specific” hypotheses in detail. If the authors decide they want to present all the detailed hypotheses, I would strongly advise presenting them using bullet points, such that it is easier for the reader to follow and consult when needed.

We have moved all hypotheses and predictions to the end of the introduction (lines 140-162) and wrote them in a more straightforward manner. We think that listing each prediction using bullet points is not necessary anymore.

Another comment I have to the Introduction concerns the ideas presented in lines 59-64:

“It is important to realise, however, that such an owner-stranger differentiation depends not only on the strength of the dog’s bond to its owners but also on the ease with which a dog accepts strangers as social partners. Stranger acceptance varies greatly across individual dogs and does so independent of the dogs’ bonds to their owners [3,10–12]. Many studies, however, do not make an effort to disentangle these two factors…” -- I wonder whether the authors aimed at disentangling this owner bond vs stranger acceptance effects in the present study. If they did, it is not currently clear how. If they did not, then maybe this specific issue can be dropped out of the Introduction.

I the present study we could directly investigate stranger’s acceptance/social support in the experimental situation thanks to the comparison with the alone condition (Experiment 2). That is, we did not only compare the behaviour of dogs in the presence of the owner and in the presence of the stranger, but compared both conditions to the alone one. Moreover, the comparison between the two groups (former shelter dogs and constant family dogs) helped us disentangling the two options. In order to make this points clearer, we revised the aims, hypotheses and predictions paragraphs at lines 137-139, 145-147 and 154-162.

Line 162-163: Please state earlier that an experimenter was always present in the room in Experiment 1, such that it is clear from early on that this was the case.

Done. It is not at lines 134-135.

Methods

Please state early on in this section (in the Experimental Design and General Procedure) when the experimenter was or was not present in the room (in Experiment 1, Experiment 2 and pre-tests).

We stated this early on at lines 135-139, but also again 199-201 in the methods section. 

Line 245 - Who was holding the dog while “she placed it on the marked spot in the experimental room”?

The owner. Added at line 218.

Lines 248-249 - Why this difference between conditions? Why didn’t the owner and the stranger entered together as well in the “owner condition”?

There was a reason only in the stranger condition for sending the owner into the room together with the stranger: had the owners stayed in the corridor from where the dog entered the room, the dogs would have likely hesitated to enter the room, thereby creating a difference between conditions (e.g. dogs showing less interest in the toys not because the owner was not in the room but because they stick to the door that separates them from the owner). There was no reason, however, to send the stranger into the room in the owner condition. On the contrary, this, unnecessarily, would have made the strangers more familiar for the dogs – an effect we wanted to avoid in the current experiment where two different persons played the role of the stranger in the two stranger conditions.

Line 259: “At the end of the trial” – The information on what determined the end of a trial should be presented already here (currently it is only presented later, on line 270)

Added at lines 228-229. 

Line 260 – Please add to where the experimenter “left the testing room together with the dog” (hallway?)

Added at line 231-232.

The info on lines 271-280 should come in line 268 right after “first in presence and then in absence of their owner”.

We moved the information regarding each trial up. Now at lines 241-250.

Lines 406-408: I think the paper benefits if this info is presented earlier (see comment above).

The presence of the experimenter in the room in Experiment 1 but not in Experiment 2 is already mentioned at lines 135-139, 199-201, 374, 385-386, hence multiple times and earlier than this point.

Results

I think there are too many graphs/tables in this section. Please think reducing the number of figures – maybe create two big figures (one for Experiment 1 and one for Experiment 2) with several panels. Moreover, I would suggest moving the tables to supplementary material.

We combined figures as suggested. Now there are only two figures for Experiment 1 and one figure for Experiment 2. Moreover, we moved the Tables to the Supporting Information and uploaded them separately from the main manuscript in the S1 File.

Discussion

My only comment to this section is that I missed seeing the present results compared/contrasted with those from Previde and Valsecchi (2007) [6, 17], the only other published study comparing attachment in shelter vs ‘family’ dogs.

Thank you for this comment. We added that indeed our in line with that previous study. We added this information at lines 531-540.

---

## [Decision Letter · Decision Letter 1]

29 Nov 2021

PONE-D-21-22971R1Secure base effect in former shelter dogs and other family dogs: strangers do no provide security in a problem-solving taskPLOS ONE

Dear Dr. Cimarelli,

Thank you for submitting your manuscript to PLOS ONE. After careful consideration, we feel that it has merit but does not fully meet PLOS ONE’s publication criteria as it currently stands. Therefore, we invite you to submit a revised version of the manuscript that addresses the points raised during the review process.

The reviewer has provided detailed suggestions for how to make the writing more stringent, and as the editor I have a few additional comments on wording, all of which is presented below.

We look forward to receiving your revised manuscript.

Kind regards,

I Anna S Olsson, Ph.D.

Academic Editor

PLOS ONE

Journal Requirements:

Additional Editor Comments:

Line 1 "do no**t**" (the letter t is missing)

Line 31 "affects" (the letter s is missing)

Line 132 Replace "[26]'s" with the author name followed by [26]

Line 168 What are the GSP guidelines? I'm not familiar with them, and googling gave me a reference to a trade term.

Line 249 Start a new paragraph at "The dog passed the pre-test"

Line 315 The term "partner" has as far as I can see not been defined previously. I could deduce that it was the person present with the dog in the test situation, but I suggest you define the term here or in the methods section, where you introduce the social conditions around the set-up

Line 486 Replace "would have had an effect" with "had an effect"

Line 489 Replace "calls attention that" with "highlight that" or "calls attention to that"

Line 593 "teacher**s**" (the letter s is missing)

Reviewers' comments:

Reviewer's Responses to Questions

**Comments to the Author**

1. If the authors have adequately addressed your comments raised in a previous round of review and you feel that this manuscript is now acceptable for publication, you may indicate that here to bypass the “Comments to the Author” section, enter your conflict of interest statement in the “Confidential to Editor” section, and submit your "Accept" recommendation.

Reviewer #1: (No Response)

2. Is the manuscript technically sound, and do the data support the conclusions?

Reviewer #1: Yes

3. Has the statistical analysis been performed appropriately and rigorously? 

Reviewer #1: I Don't Know

4. Have the authors made all data underlying the findings in their manuscript fully available?

Reviewer #1: Yes

5. Is the manuscript presented in an intelligible fashion and written in standard English?

Reviewer #1: Yes

6. Review Comments to the Author

Reviewer #1: I would like to thank the authors for the changes and clarifications provided and for the detailed responses to my comments. I think the manuscript has improved substantially. I would just like some minor issues to be addressed before I can say the paper is ready for being published. I detail my suggestions and questions in what follows.

Terms

The terms “constant family dogs” and “former shelter dogs” make the text too verbose sometimes. I understand why the authors choose these terms but, personally, for a matter of easiness of reading, I would use “family dogs” and “shelter dogs”. This is just a suggestion, and I am ok with the authors deciding to stick to the current terms.

Aims of study

I would suggest that the authors keep the presentation of the aims of the study more consistent throughout the paper. I do not mind, and actually think it is helpful for the reader, if a very similar wording (or even the same) is used.

In the end of the Introduction (lines 163-165), the authors state two aims:

- “to compare the secure base effect of owners to the social support strangers may provide dogs with”

- “to investigate how having lived in a shelter may influence stranger acceptance and bonding with the owner.”

Latter, in the beginning of the Discussion (lines 452-456), despite the authors stating the study had two main aims, I personally see three. Namely:

- “to investigate whether strangers could serve as a secure base for pet dogs”

- to investigate “to what extent they would do so less than owners would”

- “to examine whether the secure base effect provided by owners (and potentially by strangers) would differ between dogs with a past in the shelter and dogs who have always lived with the same owner.”

Line 122: I would suggested reorganizing to something like:

However, in the first study using this paradigm [26], some methodological limitations did not allow to fully compare to what extent owners and strangers could serve as a secure base for dogs. Specifically, while there was a condition in which the owner would be encouraging the dog, no corresponding condition existed for the stranger. Although results showed that dogs spent more time manipulating the object when the owner was encouraging them as compared to when the stranger was silent, and no differences were found between the silent owner and the silent stranger conditions, the possibility is still open that also an encouraging unfamiliar human could facilitate dogs’ manipulation, when compared to the owners.

Line 140-147: Here I would suggest stating which the hypothesis are for each Experiment. -- lines 140-144 refer to results to be obtained in Experiment 1 and lines 145-157 refer to results to be obtained in Experiment 2.

Line 148-162: This second part is still a bit confusing. As far as I understand, the authors are presenting three hypothesis, namely:

1. dogs might be very flexible and could be able to build new bonds easily, even when adults, resulting in no differences between constant family and former shelter dogs.

2. dogs’ re-homing past might have an effect on their social development and as such…

2.1. “Having lived in a shelter negatively affects the future development of a secure attachment bond or

2.1. “Having lived in a shelter increases the likelihood of accepting strangers as social partners

3. former shelter dogs might form a secure attachment bond with their owners comparable to the one formed by other family dogs, but be generally more anxious than constant family dogs

…but the way it is currently written does not leave it clear.

Still in Line 154: “In the first case, former shelter dogs would manipulate and explore less in the presence of the owner than other family dogs.” – what about Owner-Stranger differentiation? One would expect a less strong O-S differentiation, no?

Line 200 - I suggest introducing here the concept “coordinating experimenter” and using it every time from here on.

Line 202 - Why not referring here to the towel as well? Doesn’t this section refer to the Experimental Setup for both the pre-test and the test? Moreover, in the Materials section, the authors present the materials for both pre-test and test.

Line 218: Consider writing “held by the owner on a leash” instead of “held by the owner thanks to the leash”. Moreover, was the dog watching the coordinating experimenter from the hallway or inside the room?

Please clarify throughout the section Experimental Design and General Procedure when were the doors opened or closed. For example, in line 221 “The experimenter and the dog waited outside the testing room in the hallway” – was Door 1 opened or closed?

Line 280 - When the authors write “Similarly to the study by [26], “Manipulation” was coded whenever the dog was touching the toy with its muzzle or paw” it seems that it was just for this behavior and not the following ones. Please clarify if the remaining coded behaviors were also coded “similarly to [26]”.

Line 506-509: “It is difficult to tell whether former shelter dogs’ exploring to a similar extent in presence of the owner and of the stranger was due to lower stress when the owners were absent or more support when they were present.”

This idea is hard to follow. Maybe the problem is the fact that it is not clear whether these comparisons (“lower stress when the owners were absent” and “more support when they were present”) refers to a comparison between shelter dogs and family dogs or if it refers to a within-group comparison (i.e., owner-stranger comparison for shelter dogs). Namely, when the authors say that these dogs could be showing lower stress in the absence of the owner, does this mean lower as compared to what family dogs would show, or lower as compared to the other condition (owner present)?

It is also confusing because, as far as I understand, exploration in this paragraph seems to be interpreted both as a sign of distress and a sign of secure-base effect. In this same example, when the authors are trying to interpret the absence of differences in exploration in shelter dogs, it seems that the first hypothesis (lower stress when the owners were absent) is based on distress-related exploration and the second hypothesis (more support when they were present) is based on exploration related to a secure-base effect provided by the owner.

I truly feel this section needs clarification.

Typos:

Abstract line 31: remove “s” in “affects”.

Line 350: I believe the authors mean “Experiment 1”

Line 453: I suggest removing the “would” at the end of the sentence

7. PLOS authors have the option to publish the peer review history of their article (what does this mean?). If published, this will include your full peer review and any attached files.

Reviewer #1: **Yes: **Ana Catarina Vieira de Castro

---

## [Author Response · Author response to Decision Letter 1]

2 Dec 2021

Response to reviewers - PLOS ONE manuscript [PONE-D-21-22971R1]

Dear Editor,

Please find below the answers to each of your and the reviewers’ points. Our answers are reported in bold below each point.

Thank you very much for your work and valuable contribution to improve the present manuscript.

Kind regards,

Giulia Cimarelli, PhD

Additional Editor Comments:

Line 1 "do not" (the letter t is missing)

Corrected.

Line 31 "affects" (the letter s is missing)

Corrected.

Line 132 Replace "[26]'s" with the author name followed by [26]

Corrected.

Line 168 What are the GSP guidelines? I'm not familiar with them, and googling gave me a reference to a trade term.

Good Scientific Practice. We corrected it.

Line 249 Start a new paragraph at "The dog passed the pre-test"

Done.

Line 315 The term "partner" has as far as I can see not been defined previously. I could deduce that it was the person present with the dog in the test situation, but I suggest you define the term here or in the methods section, where you introduce the social conditions around the set-up

Added at line 273 and 289.

Line 486 Replace "would have had an effect" with "had an effect"

Corrected.

Line 489 Replace "calls attention that" with "highlight that" or "calls attention to that"

Corrected.

Line 593 "teachers" (the letter s is missing)

Corrected.

Comments to the Author

6. Review Comments to the Author

Reviewer #1: I would like to thank the authors for the changes and clarifications provided and for the detailed responses to my comments. I think the manuscript has improved substantially. I would just like some minor issues to be addressed before I can say the paper is ready for being published. I detail my suggestions and questions in what follows.

Terms

The terms “constant family dogs” and “former shelter dogs” make the text too verbose sometimes. I understand why the authors choose these terms but, personally, for a matter of easiness of reading, I would use “family dogs” and “shelter dogs”. This is just a suggestion, and I am ok with the authors deciding to stick to the current terms.

Thank you for your suggestion, we thought about it but we don’t want to confuse the reader: by simply mentioning shelter and family dogs, the risk is that the reader will interpret shelter dogs as still being host in the a shelter, while all dogs in our sample are indeed family dogs, but one group has a past in a shelter. We would like to stick with the terminology used so far, even if a but verbose.

Aims of study

I would suggest that the authors keep the presentation of the aims of the study more consistent throughout the paper. I do not mind, and actually think it is helpful for the reader, if a very similar wording (or even the same) is used.

In the end of the Introduction (lines 163-165), the authors state two aims:

- “to compare the secure base effect of owners to the social support strangers may provide dogs with”

- “to investigate how having lived in a shelter may influence stranger acceptance and bonding with the owner.”

Latter, in the beginning of the Discussion (lines 452-456), despite the authors stating the study had two main aims, I personally see three. Namely:

- “to investigate whether strangers could serve as a secure base for pet dogs”

- to investigate “to what extent they would do so less than owners would”

- “to examine whether the secure base effect provided by owners (and potentially by strangers) would differ between dogs with a past in the shelter and dogs who have always lived with the same owner.”

Thank you. We modified the end of the Introduction (Lines 165-168) to more clearly state the 3 aims of the study and be more in line with the rest of the manuscript, as the reviewer suggested: “Overall, the present study allows us to test whether strangers could serve as a secure base for pet dogs and to compare the secure base effect of strangers to the one provided by owners. Moreover, the present study aims at investigating how having lived in a shelter may influence stranger acceptance and bonding with the owner.” Moreover, we changed that the study aims are three and not two as previously stated at line 464.

Line 122: I would suggested reorganizing to something like:

However, in the first study using this paradigm [26], some methodological limitations did not allow to fully compare to what extent owners and strangers could serve as a secure base for dogs. Specifically, while there was a condition in which the owner would be encouraging the dog, no corresponding condition existed for the stranger. Although results showed that dogs spent more time manipulating the object when the owner was encouraging them as compared to when the stranger was silent, and no differences were found between the silent owner and the silent stranger conditions, the possibility is still open that also an encouraging unfamiliar human could facilitate dogs’ manipulation, when compared to the owners.

Thank you! Indeed the structure of the paragraph is simpler and clearer. We modified as the reviewer suggested. 

Line 140-147: Here I would suggest stating which the hypothesis are for each Experiment. -- lines 140-144 refer to results to be obtained in Experiment 1 and lines 145-157 refer to results to be obtained in Experiment 2.

Yes, we added in brackets to what experiment each prediction is referred to (Lines 146 and 149).

Line 148-162: This second part is still a bit confusing. As far as I understand, the authors are presenting three hypothesis, namely:

1. dogs might be very flexible and could be able to build new bonds easily, even when adults, resulting in no differences between constant family and former shelter dogs.

2. dogs’ re-homing past might have an effect on their social development and as such…

2.1. “Having lived in a shelter negatively affects the future development of a secure attachment bond or

2.1. “Having lived in a shelter increases the likelihood of accepting strangers as social partners

3. former shelter dogs might form a secure attachment bond with their owners comparable to the one formed by other family dogs, but be generally more anxious than constant family dogs

…but the way it is currently written does not leave it clear.

Thank you. We corrected that indeed the hypotheses are 3 (Line151) and we added list item letters for the in-sentence list. This should make the structuring clearer (paragraph at lines 150-164).

Still in Line 154: “In the first case, former shelter dogs would manipulate and explore less in the presence of the owner than other family dogs.” – what about Owner-Stranger differentiation? One would expect a less strong O-S differentiation, no?

Yes, we added “and a less strong owner-stranger differentiation”.

Line 200 - I suggest introducing here the concept “coordinating experimenter” and using it every time from here on.

Thanks for you suggestion, we added it and used it throughout the text.

Line 202 - Why not referring here to the towel as well? Doesn’t this section refer to the Experimental Setup for both the pre-test and the test? Moreover, in the Materials section, the authors present the materials for both pre-test and test.

Yes, we added it at line 206.

Line 218: Consider writing “held by the owner on a leash” instead of “held by the owner thanks to the leash”. 

Corrected.

Moreover, was the dog watching the coordinating experimenter from the hallway or inside the room?

From the hallway, we added it at line 223.

Please clarify throughout the section Experimental Design and General Procedure when were the doors opened or closed. For example, in line 221 “The experimenter and the dog waited outside the testing room in the hallway” – was Door 1 opened or closed?

Closed. We added details regarding whether the door was open or closed throughout the whole section.

Line 280 - When the authors write “Similarly to the study by [26], “Manipulation” was coded whenever the dog was touching the toy with its muzzle or paw” it seems that it was just for this behavior and not the following ones. Please clarify if the remaining coded behaviors were also coded “similarly to [26]”.

All behaviours were coded following [26]. We specified this at line 291 and we used semicolons instead of colons to list the different variables.

Line 506-509: “It is difficult to tell whether former shelter dogs’ exploring to a similar extent in presence of the owner and of the stranger was due to lower stress when the owners were absent or more support when they were present.”

This idea is hard to follow. Maybe the problem is the fact that it is not clear whether these comparisons (“lower stress when the owners were absent” and “more support when they were present”) refers to a comparison between shelter dogs and family dogs or if it refers to a within-group comparison (i.e., owner-stranger comparison for shelter dogs). Namely, when the authors say that these dogs could be showing lower stress in the absence of the owner, does this mean lower as compared to what family dogs would show, or lower as compared to the other condition (owner present)? It is also confusing because, as far as I understand, exploration in this paragraph seems to be interpreted both as a sign of distress and a sign of secure-base effect. In this same example, when the authors are trying to interpret the absence of differences in exploration in shelter dogs, it seems that the first hypothesis (lower stress when the owners were absent) is based on distress-related exploration and the second hypothesis (more support when they were present) is based on exploration related to a secure-base effect provided by the owner.

I truly feel this section needs clarification.

Only within the former shelter group. And yes, exploration could be either a sign of distress or of a secure base effect. We modified the sentence to make this clearer “If exploration was a sign of distress also in the case of former shelter dogs, then a similar duration of exploration in the presence of the owner and of the stranger might be either due to low stress when the owner is absent (stranger condition) or to received support when the owner is present (owner condition).” We additionally simplified the rest of the paragraph (lines 515-527).

Typos:

Abstract line 31: remove “s” in “affects”.

Corrected.

Line 350: I believe the authors mean “Experiment 1”

Corrected.

Line 453: I suggest removing the “would” at the end of the sentence

Corrected.

---

## [Decision Letter · Decision Letter 2]

10 Dec 2021

Secure base effect in former shelter dogs and other family dogs: strangers do not provide security in a problem-solving task

PONE-D-21-22971R2

Dear Dr. Cimarelli,

We’re pleased to inform you that your manuscript has been judged scientifically suitable for publication and will be formally accepted for publication once it meets all outstanding technical requirements.

Kind regards,

I Anna S Olsson, Ph.D.

Academic Editor

PLOS ONE

Additional Editor Comments (optional):

Reviewers' comments:

Reviewer's Responses to Questions

**Comments to the Author**

1. If the authors have adequately addressed your comments raised in a previous round of review and you feel that this manuscript is now acceptable for publication, you may indicate that here to bypass the “Comments to the Author” section, enter your conflict of interest statement in the “Confidential to Editor” section, and submit your "Accept" recommendation.

Reviewer #1: All comments have been addressed

2. Is the manuscript technically sound, and do the data support the conclusions?

Reviewer #1: Yes

3. Has the statistical analysis been performed appropriately and rigorously? 

Reviewer #1: I Don't Know

4. Have the authors made all data underlying the findings in their manuscript fully available?

Reviewer #1: Yes

5. Is the manuscript presented in an intelligible fashion and written in standard English?

Reviewer #1: Yes

6. Review Comments to the Author

Reviewer #1: (No Response)

7. PLOS authors have the option to publish the peer review history of their article (what does this mean?). If published, this will include your full peer review and any attached files.

Reviewer #1: **Yes: **Ana Catarina Vieira de Castro

---

## [Editor Report · Acceptance letter]

13 Dec 2021

PONE-D-21-22971R2 

Secure base effect in former shelter dogs and other family dogs: strangers do not provide security in a problem-solving task 

Dear Dr. Cimarelli:

I'm pleased to inform you that your manuscript has been deemed suitable for publication in PLOS ONE. Congratulations! Your manuscript is now with our production department. 

Kind regards, 

on behalf of

Dr. I Anna S Olsson 

Academic Editor

PLOS ONE